# First-line systemic treatment strategies for unresectable hepatocellular carcinoma: A cost-effectiveness analysis

Liting wang[1], Ye Peng[1☯], Shuxia Qin[1☯], Xiaomin Wan[1☯], Xiaohui Zeng[2☯], Sini Li[3], Qiao Liu[1]*, Chongqing Tan[1]*

1 Department of Pharmacy, the Second Xiangya Hospital of Central South University, Changsha, Hunan, China, 2 PET-CT Center, the Second Xiangya Hospital of Central South University, Changsha, Hunan, China, 3 Faculty of Medicine, School of Health and Related Research, Dentistry and Health, University of Sheffield, Sheffield, United Kingdom

☯ These authors contributed equally to this work.
* liuqiao6767@csu.edu.cn (QL); tanchongqing@csu.edu.cn (CT)

**Data Availability Statement:** All relevant data are within the paper.

**Funding:** The work was supported by grants from the National Natural Science Foundation of China

## Abstract

### Background

Oral multikinase inhibitors and immune checkpoint inhibitors (ICIs) are effective for treating advanced hepatocellular carcinoma (aHCC) but may increase cost. This study compared the cost-effectiveness of oral multikinase inhibitors and ICIs in the first-line treatment of patients with aHCC.

### Methods

A three-state Markov model was established to study the cost-effectiveness of drug treatment from the perspective of Chinese payers. The key outcomes in this study were total cost, quality-adjusted life years (QALYs), and the incremental cost-effectiveness ratio (ICER).

### Results

The total costs and QALYs of sorafenib, sunitinib, donafenib, lenvatinib, sorafenib plus erlotinib, linifanib, brivanib, sintilimab plus IBI305, and atezolizumab plus bevacizumab were $9070 and 0.25, $9362 and 0.78, $33,814 and 0.45, $49,120 and 0.83, $63,064 and 0.81, $74,814 and 0.82, $81,995 and 0.82, $74083 and 0.85, and $104,188 and 0.84, respectively. The drug regimen with the lowest ICER was sunitinib ($551 per QALY), followed by lenvatinib ($68,869 per QALY). For oral multikinase inhibitors, the ICER of lenvatinib, sorafenib plus erlotinib, linifanib and brivanib compared with sunitinib was $779576, $1534,347, $1768,971, and $1963,064, respectively. For ICIs, sintilimab plus IBI305 is more cost effective than atezolizumab plus bevacizumab. The model was most sensitive to the price of sorafenib, the utility of PD, and the price of second-line drugs.

(grant numbers: 82073818), 2, The Fundamental Research Funds for the Central Universities of Central South University (grant numbers 2022ZZTS0282), 3, Supported by Hunan Provincial Innovation Foundation For Postgraduate (grant numbers: CX20220364).

**Competing interests:** The authors have declared that no competing interests exist.

## Conclusion

For oral multikinase inhibitors, the order of possible treatment options is sunitinib > lenvatinib > sorafenib plus erlotinib > linifanib > brivanib > donafenib. For ICIs, the order of possible treatment options is sintilimab plus IBI305 > atezolizumab plus bevacizumab.

## Introduction

Liver cancer is the sixth most common malignancy and the third leading cause of cancer-related mortality worldwide [1]. Hepatocellular carcinoma (HCC) represents 75%-80% of all liver cancers [1], and the vast majority of HCC is caused by hepatitis B virus (HBV) infection. China accounts for approximately half of the world's new HBV infections [2]. Most HCC cases are diagnosed at an advanced stage with a 5-year survival of less than 10% [3]. In addition, local resection is not a good option for patients with advanced liver cancer, and therefore, very few patients are eligible for transplantation [4].

Sorafenib is the first oral multitarget tyrosine kinase inhibitor (TKI) approved by China in 2007 for the systematic treatment of unresectable advanced HCC (aHCC) [5]. The approval of sorafenib was a response to the positive results from two pivotal phase III trials, i.e., the SHARP and NCT00492752 trials [6,7], which led to the era of systemic pharmacotherapy for unresectable aHCC [8]. Subsequently, many drugs have been approved for the first-line treatment of unresectable advanced HCC, including oral multikinase inhibitors sunitinib [9], brivanib [10], erlotinib [11], linifanib [12], lenvatinib [13] and donafenib [14], as well as immune checkpoint inhibitors (ICIs) atezolizumab [15] and sintilimab [16]. With the emergence of these novel drugs, physicians and patients face difficulties in determining which is preferable.

Recently, a network meta-analysis compared the efficacy and safety of these approved first-line systemic treatment strategies (including the drugs mentioned above) [17]. However, the generalizability of their findings to clinical practice may be limited, in which treatment decision-making needs to juggle both cost and efficacy. Therefore, there is a need to conduct a cost-effectiveness analysis to decide the priority of first-line systemic treatment strategies, especially in resource-poor countries such as China [18]. Moreover, we used a Markov model to build a cost-effectiveness model from the Chinese health care system to rank first-line treatments for aHCC.

## Materials and methods

### Analytical overview

A hypothetical cohort of patients with unresectable aHCC who had not previously received systemic therapy was assumed in the model, which mirrored the participants recruited in the SHARP and NCT00492752 trials [6,7]. The treatment strategies considered in this analysis included the following nine first-line systematic treatments: sunitinib, donafenib, lenvatinib, sorafenib plus erlotinib, sintilimab plus IBI305, linifanib, brivanib, atezolizumab plus bevacizumab, and sorafenib. A three-state Markov model consisting of progression-free survival (PFS), progressed disease (PD) and death was constructed to compare the cost-effectiveness of the other eight drugs versus sorafenib in unresectable aHCC (Fig 1). We used a 21-day cycle length and a 10-year time horizon to output the total cost, life-years (LYs), and quality-adjusted life years (QALY) associated with each treatment strategy. The cost-effectiveness was measured by using the incremental cost–benefit ratio (ICER), with an ICER lower than the

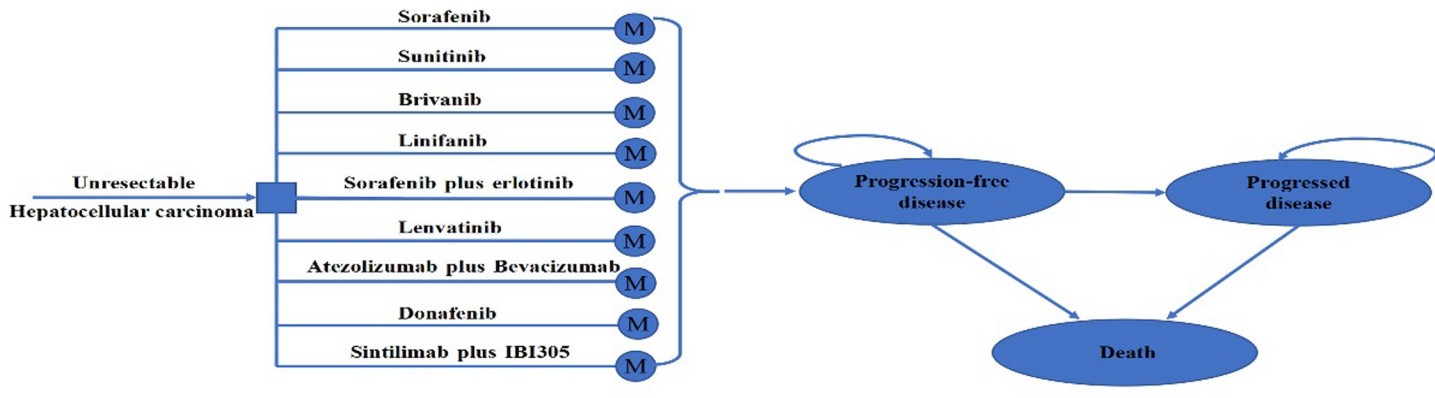

**Fig 1. Model structure of a Markov model combining the decision tree.** M, Markov.

willingness-to-pay (WTP) of $37,654.50 (defined as 3 times China's GDP per capita in 2021) [19]. All costs were expressed in 2022 US dollars using the exchange rate of 1 US dollar equivalent to 6.31 Chinese yuan (March 2022). A 5% annual discount rate was used for both costs and effectiveness [20]. The model was established and analyzed by TreeAge Pro (TreeAge Software, Williamstown, MA) and R software version 3.6.1 (https://www.r-project.org/).

## Clinical data inputs

The probability of PFS status transition was estimated from the PFS curve reported in the SHARP and NCT00492752 trials [6,7], the transition probability from the PFS state to the PD state was calculated using the difference between the OS and PFS curves of the clinical trial, and the final probability of death was obtained by subtracting the area under the OS curve of the clinical trial from the PFS and OS data of sorafenib patients derived from the SHARP and NCT00492752 trial results [6,7]. Data points were extracted from OS curves and PFS curves reported in the SHARP and NCT00492752 trials using GetData Graphic Digitizer, version 2.26, and then the algorithm proposed by Guyot et al. was used to generate time ranges outside the model [21]. Finally, Weibull, exponential, log-logistic, log-normal, and Gompertz survival functions were fitted, and the last survival function selected according to Akaike information criterion (AIC) was log-logistic. The survival function for other drugs was adjusted using HR

**Table 1. Dosage of treatment regimen.**

| treatment regimen | Dosage |
|---|---|
| First-line treatment | |
| Sorafenib | 400mg twice a day |
| Sunitinib | 12.5mg once a day |
| Donafenib | 200mg twice daily |
| Lenvatinib | 12 mg(bodyweight ≥60 kg) or 8mg(bodyweight < 60 kg) once daily |
| Sorafenib plus Erlotinib | 400mg twice daily and erlotinib 150mg once daily |
| Sintilimab plus (IBI305) | 200 mg sintilimab and 15 mg/kg bevacizumab biosimilar every 3 weeks |
| Linifanib | 17.5mg once daily |
| Brivanib | 800mg once daily |
| Atezolizumab plus Bevacizumab | 1200 mg atezolizumab and 15 mg/kg bevacizumab every 3 weeks |
| Second-line treatment | |
| Camrelizumab plus apatinib | 200 mg Camrelizumab every 2 weeks and 250 mg apatinib once daily |

from a network meta-analysis [17]. See S1 Table for original data. The subsequent therapy strategies after disease progression were based on guideline recommendations [22]. Key clinical inputs are shown in S2 Table.

## Cost and utility inputs

Only direct health care costs were covered in this analysis, including drug costs, testing costs and AE costs (S2 Table). All costs were reported in 2022 US dollars and adjusted to 2022 values based on the consumer Price Index [23].

The study compared nine treatment strategies. The dosage of the medication regimen is shown in Table 1. To calculate the dosage of lenvatinib, bevacizumab and IBI305, we modeled the baseline patients as weighing 60 kg [2]. Patients in all groups were assumed to have continued first-line treatment until the disease progressed or unacceptable toxicity occurred. After disease progression, patients were allowed to receive camrelizumab plus apatinib as second-line treatment according to Chinese guidelines [23]. Based on a second-line treatment trial for unresectable hepatocellular carcinoma [24], patients received intravenous camrelizumab 200 mg every 2 weeks plus oral apatinib 250 mg daily. All the information is shown in Table 1.

QALYs were measured as a weighted aggregate of health utilities over time. In this analysis, the utility scores assigned to the PFS and PD health states were 0.760 and 0.680 [25], respectively. Moreover, experiencing grade 3 or 4 adverse events during each first-line systematic treatment was considered a decrement in health utilities [26]; this information is shown in S2 Table.

## Base-case analysis

Total expected costs, LYs, QALYs and ICERs were estimated for each first-line systematic treatment. To identify the sensitive parameters that affect our model economic outcomes, we performed both 1-way sensitivity analyses and probabilistic sensitivity analysis. During 1-way sensitivity analyses, each parameter was tested within the range of plus or minus 20% of the baseline value or within the plausible ranges from published literature (S2 Table). During probabilistic sensitivity analysis, 10,000 iterations of Monte Carlo simulations were performed by setting appropriate distributions for each parameter. We selected beta distributions for probability, proportion, and preference value parameters, log-normal distributions for the HRs, and gamma distributions for the cost parameters. The cost-effective acceptability curve indicates the possibility of being considered cost-effective at different levels of WTP.

## Ethics statement

This study was based on a literature review and modeling techniques; this study did not require approval by an institutional research ethics board.

## Results

### Base-case analysis

Table 2 summarizes the base-case results. The atezolizumab plus bevacizumab group had the highest total cost (US $104,188), and the sorafenib group had the lowest total cost (US $9,362). The QALYs obtained by atezolizumab plus bevacizumab, sintilimab plus IBI305, brivanib, linifanib, sorafenib plus erlotinib, lenvatinib, donafenib and sunitinib relative to the sorafenib treatment group were 0.84, 0.85, 0.82, 0.82, 0.81, 0.83, 0.45 and 0.78, respectively, increasing by 0.59, 0.56, 0.57, 0.59, 0.57, 0.58, 0.20 and 0.53 QALYs. When patients received oral multikinase inhibitors as first-line systematic treatments, compared with sorafenib, the ICER for lenvatinib, sorafenib plus erlotinib, linifanib, brivanib versus sunitinib were $779,576, $1534,347,

**Table 2. Base case results.**

| Treatment strategies | Cost | incrC | QALY | incrE | Scheme comparison | ICER |
|---|---|---|---|---|---|---|
| Sorafenib | 9070 | / | 0.25 | / | / | |
| Sunitinib | 9362 | 292 | 0.78 | 0.53 | Sorafenib | 551 |
| Donafenib | 33814 | 24744 | 0.45 | 0.20 | Sorafenib | 121059 |
| Lenvatinib | 49120 | 40050 | 0.83 | 0.58 | Sorafenib | 68869 |
| Sorafenib Plus Erlotinib | 63064 | 53994 | 0.81 | 0.57 | Sorafenib | 95545 |
| Linifanib | 74814 | 65744 | 0.82 | 0.59 | Sorafenib | 115760 |
| Brivanib | 81995 | 72925 | 0.82 | 0.57 | Sorafenib | 128527 |
| Sintilimab plus (IBI305) | 74083 | 65013 | 0.85 | 0.56 | Sorafenib | 115760 |
| Atezolizumab plus Bevacizumab | 104188 | 95118 | 0.84 | 0.59 | Sorafenib | 160049 |

incrC: Incremental cost, QALYs: Quality-adjusted-life-years; incrE: Incremental effect; ICER: Incremental cost-effectiveness ratio.

$176,8971 and $1963,064 per QALY, respectively. When patients received immune checkpoint inhibitors (ICIs) as first-line systematic treatment, atezolizumab plus bevacizumab was dominated by sintilimab plus bevacizumab IBI305. Detailed data are shown in Table 3.

## Sensitivity analysis

The results of the one-way sensitivity analysis showed that for sunitinib versus sorafenib, the ICER was particularly sensitive to the price of sorafenib; for donafenib, sorafenib plus erlotinib, linifanib, brivanib, sintilimab plus IBI305 and atezolizumab plus bevacizumab versus sorafenib, the utility of PD was the most sensitive parameter; for lenvatinib versus sorafenib, the price of second-line drugs was the most sensitive parameter; for sunitinib, sorafenib plus erlotinib and brivanib versus lenvatinib, the utility of PD was the most sensitive parameter; and for donafenib versus lenvatinib, the price of second-line drugs was the most sensitive parameter. Other sensitive parameters are shown in S1 Fig. With the increase in the WTP thresholds, the possibility of cost-effectiveness among different treatment drugs increased, but the cost-effectiveness of all drug treatment regimens except sorafenib was less than 50%, as shown in Fig 2.

**Table 3. Results of oral multikinase inhibitor group and ICI group.**

| Drug Name | Total cost($) | QALYs | ICER (per QALY) |
|---|---|---|---|
| oral multikinase inhibitors | | | |
| Sunitinib | 9362 | 0.78 | / |
| Lenvatinib | 49120 | 0.83 | 7795762 |
| Sorafenib Plus Erlotinib | 63064 | 0.81 | 1534347 |
| Linifanib | 74814 | 0.82 | 1768971 |
| Brivanib | 81995 | 0.82 | 1963064 |
| Donafenib | 33814 | 0.45 | dominated |
| immune checkpoint inhibitors (ICI) | | | |
| Sintilimab plus IBI305 | 74083 | 0.85 | / |
| Atezolizumab plus Bevacizumab | 104188 | 0.84 | dominated |

QALYs, quality-adjusted life-years. ICER, incremental cost-effectiveness ratio.

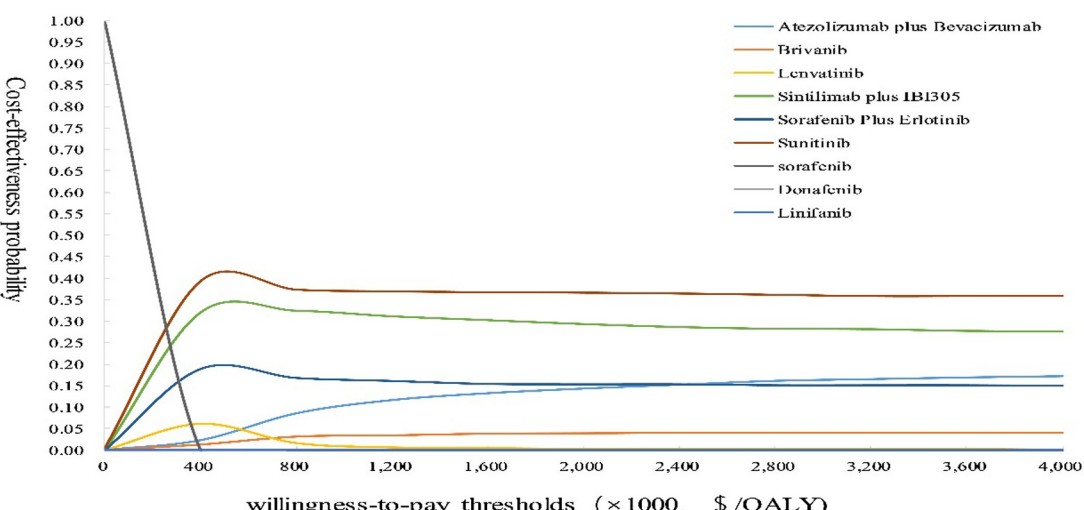

**Fig 2. Cost-effectiveness acceptability curves.** QALYs, quality-adjusted life-years.

## Discussion

We used the three-state Markov model to analyze the cost-effectiveness of eight first-line treatment plans. The drugs of the eight treatment plans are mainly oral multikinase inhibitors and immune checkpoint inhibitors. Compared with standard sorafenib treatment, all other first-line treatments were superior in improving survival; among them, the sunitinib treatment plan is a cost-effective choice, with an ICER of $551/QALYs below the WTP threshold of $37,654.50/QALYs. China also began negotiating with pharmaceutical companies over the price of cancer drugs after establishing the National Medical Safety Administration (NHSA) in May 2018. Among them, sunitinib, lenvatinib and sintilimab entered the NHSA, and we look forward to more drugs. For oral multikinase inhibitors, our secondary study found that the QALY value of lenvatinib was 0.83, which was the largest, and this first-line treatment plan certainly had the best long-term efficacy. Moreover, the indication for lenvatinib to be included in the medical insurance list was advanced hepatocellular carcinoma, which could reduce the economic burden of many patients. We assumed that the patient's weight was 60 kg and the dose was 8 mg and did not consider the 12 mg dose for the patient's weight over 60 kg. Hongfu Cai et al. found that both treatment regimens weighing less than 60 kg and more than 60 kg were cost-effective, and sensitivity analysis showed that weight was not a major influencing factor [27]. Therefore, lenvatinib has a greater chance of being more cost-effective than the other four drugs. In the REFLECT clinical trial, lenvatinib was reported to be more effective than sunitinib, brivanib, linifanib, and erlotinib plus sorafenib [13]. Hongfu Cai et al. proved the cost-effectiveness of lenvatinib versus sorafenib from the perspective of China, and Brandon M Meyers et al. demonstrated the cost-effectiveness of lenvatinib versus sorafenib from a Canadian perspective [28]. Christopher Sherrow et al. proposed that lenvatinib is the most cost-effective first-line treatment when only oral therapy is considered in the sequencing of systematic treatment options for aHCC [28], which is consistent with our findings.

When comparing the two ICI schemes, sintilimab plus IBI305 has a greater chance of being cost-effective relative to atezolizumab plus bevacizumab. Feng Wen et al. found that atezolizumab plus bevacizumab was not cost-effective from the Perspective of China [25]. Although the FDA's recommendation to use atezolizumab plus bevacizumab in patients with aHCC who have not previously received systemic treatment has ushered in a new era of systemic

treatment for hepatocellular carcinoma [29], the high cost is an issue that cannot be ignored, especially in developing countries. Ye Peng et al. and Ting Zhou et al. demonstrated the cost-effectiveness of sintilimab plus bevacizumab biosimilar from a Chinese perspective for aHCC patients [4,30]. Similarly, compared with other ICIs, the FDA of the United States and National Medical Products Administration (NMPA) of China also confirmed that sintilimab has similar antitumor effects, better safety and obvious economic advantages, which is a valuable finding for Chinese patients. Therefore, there are discrepancies between the results of the sintilimab study and those of our article, which may be due to differences in second-line regimens. From our study, we know that the price of domestic anticancer drugs is much lower than that of imported anticancer drugs, so cost-effectiveness analysis is of great significance for reducing national health expenditure. As seen from the above, domestic sintilimab has been included in the National Reimbursement Drug List (NRDL), but the indication is Hodgkin's lymphoma. With the success of the phase III clinical trial Orient-32, we believe that the indication for unresectable advanced HCC will soon be included in the NRDL.

Notably, as shown in ICER values (Table 2), only sunitinib was cost-effective according to China's WTP threshold in the range of $38,498.89/QALY; no other systematic drug regimen has been deemed cost-effective. The calculated ICER values all exceeded this threshold. The main reason is that the cost may be affected by the second-line treatment, which can be seen from the results of one-way sensitivity analysis. Apart from sunitinib, the biggest factors affecting the ICER of other treatment drugs are the cost of second-line drugs and the utility of PD. Nassir A. Azimi et al. found that if the ICER is greater than an increase of $166,000/QALY, researchers are generally opposed to implementing this treatment regimen. If ICER is in the range of $61,500 - $166,000/QALY, cost-effectiveness is ambiguous [31], so researchers come to different conclusions in this range. Our study clearly outlines the cost of each first-line drug treatment regimen to select the most cost-effective regimen for Chinese patients.

There are some limitations to our study. First, safety and efficacy data used in our study were extracted from published trials, and any biases in these trials may inevitably affect our results. Second, the potential heterogeneity across different patient populations was not considered in the model, resulting in no subgroup analysis. We did not consider effective biomarkers, which can predict the outcome of immunotherapy and are critical in the logical context of optimal cost-effectiveness. Third, the assumptions of the data also limited our analysis for second-line treatment. Due to the lack of real-world data validation, we assume that camrelizumab plus apatinib system drug therapy recommended by Chinese hepatocellular carcinoma guidelines is considered. Because this article is from the perspective of China, hoping to facilitate Chinese patients, there is no denying that it will cause deviation in clinical practice, but the focus of our study is the first-line drug treatment, the unification of second-line treatment, and a better understanding of the differences between first-line drug treatments. We also assume that the threshold for sensitivity analysis is 20% above or below, which may affect the reality when interpreting uncertainty; however, in similar studies, this range was shown to be acceptable [32]. Finally, our analysis does not address the effects of different payment options, which is a more practical issue for most policy-makers and patients

## Conclusions

Our study demonstrates the possibility of cost-effectiveness of different drug regimens and provides different treatment ideas for advanced HCC patients in China.

## Supporting information

**S1 Fig.** One-way sensitivity analyses of atezolizumab plus bevacizumab (A), brivanib (B), linifanib (C), sorafenib plus erlotinib (D), donafenib(E), lenvatinib(F), sintilimab plus IBI305(G), sunitinib(H) in comparison with sorafenib, linifanib(I), donafenib(J), sorafenib plus erlotinib(K) in comparison with lenvatinib, sintilimab plus IBI305 vs. atezolizumab plus bevacizumab(M). (DOCX)

**S1 Table. The original data of the network meta.**
(DOCX)

**S2 Table. Key model inputs.**
(DOCX)

## Author Contributions

**Investigation:** Ye Peng.

**Resources:** Qiao Liu.

**Software:** Shuxia Qin.

**Supervision:** Xiaohui Zeng, Sini Li, Chongqing Tan.

**Validation:** Xiaomin Wan.

**Writing – original draft:** Liting wang.

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
