## [Decision Letter · Decision Letter 0]

1 Feb 2023

PONE-D-22-33761First-Line Systemic Treatment Strategies for unresectable hepatocellular carcinoma : a cost -effectiveness analysisPLOS ONE

Dear Dr. Liu,

Thank you for submitting your manuscript to PLOS ONE. After careful consideration, we feel that it has merit but does not fully meet PLOS ONE’s publication criteria as it currently stands. Therefore, we invite you to submit a revised version of the manuscript that addresses the points raised during the review process. We recommend to follow the suggestions made by the reviewers. In particular we suggest to revise the English language and to modify the "conclusions" section of the abstract.

We look forward to receiving your revised manuscript.

Kind regards,

Daniele Ugo Tari, M.D.

Academic Editor

PLOS ONE

Journal Requirements:

"1、The work was supported by grants from the National Natural Science Foundation of China (grant numbers: 82073818)

2、The Fundamental Research Funds for the Central Universities of Central South University（grant numbers:2022ZZTS0282）

3、Supported by Hunan Provincial Innovation Foundation For Postgraduate （grant numbers: CX20220364）"

"YES - Specify the role(s) played."

"NO authors have competing interests

Enter: The authors have declared that no competing interests exist."

6. PLOS requires an ORCID iD for the corresponding author in Editorial Manager on papers submitted after December 6th, 2016. Please ensure that you have an ORCID iD and that it is validated in Editorial Manager. To do this, go to ‘Update my Information’ (in the upper left-hand corner of the main menu), and click on the Fetch/Validate link next to the ORCID field. This will take you to the ORCID site and allow you to create a new iD or authenticate a pre-existing iD in Editorial Manager. Please see the following video for instructions on linking an ORCID iD to your Editorial Manager account: https://www.youtube.com/watch?v=_xcclfuvtxQ.

8. We note you have included a table to which you do not refer in the text of your manuscript. Please ensure that you refer to Table 4 in your text; if accepted, production will need this reference to link the reader to the Table.

Reviewers' comments:

Reviewer's Responses to Questions

**Comments to the Author**

1. Is the manuscript technically sound, and do the data support the conclusions?

Reviewer #1: Yes

2. Has the statistical analysis been performed appropriately and rigorously? 

Reviewer #1: Yes

3. Have the authors made all data underlying the findings in their manuscript fully available?

Reviewer #1: Yes

4. Is the manuscript presented in an intelligible fashion and written in standard English?

Reviewer #1: No

5. Review Comments to the Author

Reviewer #1: Dear Author,

I congratulate you on the interesting topic provided.

The statistical analysis is rigorous and appropriate.

Your research provides useful data about the possibility to introduce new drug regimens in China with higher cost-effectiveness.

I suggest a revision of the English language since there are some typos and some sentences are unclear.

In particular I suggest to modify the "conclusion" section in the abstract avoiding the use of ">".

Consequently I think that the paper could be accepted after these minor revisions.

Sincerely,

6. PLOS authors have the option to publish the peer review history of their article (what does this mean?). If published, this will include your full peer review and any attached files.

Reviewer #1: No

---

## [Author Response · Author response to Decision Letter 0]

15 Mar 2023

Responses to Reviewer ：

4. Is the manuscript presented in an intelligible fashion and written in standard English?

Answer the question:I am very sorry that there are many grammatical mistakes in the article. The language of the article has been modified by English majors and the language has been greatly improved.

---

## [Editor Report · Decision Letter 1]

20 Mar 2023

First-Line Systemic Treatment Strategies for unresectable hepatocellular carcinoma : a cost -effectiveness analysis

PONE-D-22-33761R1

Dear Dr. Liu,

We’re pleased to inform you that your manuscript has been judged scientifically suitable for publication and will be formally accepted for publication once it meets all outstanding technical requirements.

Kind regards,

Daniele Ugo Tari, M.D.

Academic Editor

PLOS ONE

Additional Editor Comments (optional):

Dear Authors,

The paper has been improved according to reviewer's suggestions.

Consequently, I think that it may be accepted for publication in present form.

Sincerely,
---

## [Editor Report · Acceptance letter]

6 Apr 2023

PONE-D-22-33761R1 

First-line systemic treatment strategies for unresectable hepatocellular carcinoma: A cost-effectiveness analysis 

Dear Dr. Liu:

I'm pleased to inform you that your manuscript has been deemed suitable for publication in PLOS ONE. Congratulations! Your manuscript is now with our production department. 

Kind regards, 

on behalf of

Dr. Daniele Ugo Tari 

Academic Editor

PLOS ONE